# Amygdala neural activity reflects spatial attention towards stimuli promising reward or threatening punishment

Christopher J Peck[1], C Daniel Salzman[1,2,3,4,5]*

[1]Department of Neuroscience, Columbia University, New York, United States; [2]Department of Psychiatry, Columbia University, New York, United States; [3]Kavli Institute for Brain Sciences, Columbia University, New York, United States; [4]W M Keck Center on Brain Plasticity and Cognition, Columbia University, New York, United States; [5]New York State Psychiatric Institute, New York, United States

**Abstract** Humans and other animals routinely identify and attend to sensory stimuli so as to rapidly acquire rewards or avoid aversive experiences. Emotional arousal, a process mediated by the amygdala, can enhance attention to stimuli in a non-spatial manner. However, amygdala neural activity was recently shown to encode spatial information about reward-predictive stimuli, and to correlate with spatial attention allocation. If representing the motivational significance of sensory stimuli within a spatial framework reflects a general principle of amygdala function, then spatially selective neural responses should also be elicited by sensory stimuli threatening aversive events. Recordings from amygdala neurons were therefore obtained while monkeys directed spatial attention towards stimuli promising reward or threatening punishment. Neural responses encoded spatial information similarly for stimuli associated with both valences of reinforcement, and responses reflected spatial attention allocation. The amygdala therefore may act to enhance spatial attention to sensory stimuli associated with rewarding or aversive experiences.

*For correspondence: cds2005@columbia.edu

Competing interests: The authors declare that no competing interests exist.

## Introduction

Inherently emotional stimuli and stimuli that have acquired emotional meaning through learning can enhance spatial attention (*Armony and Dolan, 2002*; *Maunsell, 2004*; *Anderson, 2005*; *Phelps et al., 2006*; *Peck et al., 2009*). Enhanced spatial attention improves sensory processing of behaviorally relevant stimuli (*Anderson et al., 2011*) and quickens purposeful actions based on these stimuli (*Posner, 1980*). Subjects must not only register the emotional significance of a stimulus to enhance spatial attention, but they also must locate the stimulus. For example, in the wild, carnivores must identify potential predators and prey, register their emotional or motivational significance, and then locate the predator or prey if they want to mount an attack or retreat. The brain must therefore combine information about the motivational significance and location of stimuli to enhance spatial attention and facilitate actions.

The amygdala, a brain area traditionally linked to emotional processes (*LeDoux, 2000*; *Phelps and LeDoux, 2005*; *Murray, 2007*; *Pessoa and Adolphs, 2010*), may play an important role in enhancing spatial attention to emotional stimuli. A substantial body of literature has documented the amygdala's importance in processing both aversive (*Campeau and Davis, 1995*; *Quirk et al., 1995*) and appetitive stimuli (*Sanghera et al., 1979*; *Nishijo et al., 1988*; *Holland and Gallagher, 1993*; *Schoenbaum et al., 1998*; *Baxter and Murray, 2002*; *Carelli et al., 2003*; *Sugase-Miyamoto and Richmond, 2005*; *Ambroggi et al., 2008*; *Tye et al., 2008*; *Shabel and Janak, 2009*; *Bermudez and Schultz, 2010*; *Jenison et al., 2011*). Recent studies have compared amygdala neural responses to the same conditioned

**eLife digest** In our everyday lives, we are surrounded by stimuli that compete for our attention. However, the brain pays more attention to some stimuli—such as those that signal rewards or warn of potential threats—than to others. These stimuli receive extra attention because they activate a structure deep within the brain called the amygdala.

The amygdala, which is named after the Greek word for 'almond' owing to its shape, receives input from the sensory areas of the brain, and sends output to the hypothalamus, which controls the body's stress response, and other structures. While the role of the amygdala in signaling the presence of threats or rewards has been recognized for many years, recent studies have suggested that the amygdala also signals the location of potential rewards.

Using electrodes to record electrical signals from the amygdala of awake monkeys, Peck and Salzman now show that it also alerts animals to the location of potential threats. In response to cues appearing on a screen, the monkeys moved their eyes in a way that either earned them a reward (fruit juice), or enabled them to avoid an annoying puff of air. As expected, amygdala neurons responded to cues that predicted the reward and also to cues that signaled the air puff. Surprisingly, however, the neurons also varied their responses according to the location of the cues.

This dual function of the amygdala—signaling both the presence and the location of stimuli that predict rewards and threats—helps animals to plan whether or not they will approach or avoid a stimulus. Moreover, given that some of the most salient reward and punishment cues that we encounter today are facial expressions—which elicit strong amygdala responses—it could also provide clues to disorders of social functioning such as autism.

stimulus (CS) when paired with a rewarding or aversive unconditioned stimulus (US); neurons often responded differentially to the CS depending on whether it predicted a positive or negative outcome (*Paton et al., 2006*; *Belova et al., 2007*, *2008*; *Morrison et al., 2011*) suggestive of a valence-specific coding scheme where neurons respond to stimuli on a 'good-to-bad' scale. Further, different populations of neurons fired more for either reward- or punishment-predicting stimuli, raising the possibility that the amygdala contains distinct networks for processing stimuli possessing appetitive and aversive associations (*Zhang et al., 2013*). Neural signals encoding valence are likely critical for a range of cognitive and behavioral functions where adaptive responses differ fundamentally depending upon valence, such as approach and defensive or avoidance behaviors, economic choice behavior, and many psychophysiological responses that are known to be valence-specific (e.g. the startle response) (*Lang and Davis, 2006*). Indeed, a recent study has now documented that distinct appetitive and aversive circuits in the amygdala are causally related to valence-specific behavior (*Redondo et al., 2014*).

Valence alone cannot describe all emotions, as both positive and negative emotional experiences can also vary in intensity. Emotional intensity may be related to processes like arousal, which can be triggered by stimuli of both valences and be characterized quantitatively by psychophysiological measures (*Lang et al., 1993*). Prior studies indicate that the firing rates of amygdala neurons are correlated in some circumstances with valence-nonspecific aspects of conditioned (*Shabel and Janak, 2009*) and unconditioned stimuli (*Belova et al., 2007*), suggesting that the amygdala could modulate arousal or related processes. Nearly all previous studies have assumed that if the amygdala modulates valence-nonspecific processes, it does so in a non-spatial manner (*Holland and Gallagher, 1999*; *Maddux et al., 2007*).

Recent work, however, has provided a new conceptual framework for understanding how the amygdala might modulate valence-nonspecific processes, as neural activity in the amygdala has been linked to spatial attention. Amygdala neurons encode information about both the spatial location and reward association of visual stimuli (*Peck et al., 2013*; *Peck and Salzman, 2014*), and the maintenance of coordinated amygdala signals representing space and reward is task dependent (*Peck et al., In press*). The encoding of space and reward in the primate amygdala has now also been confirmed by human neuroimaging data (*Ousdal et al., 2014*). Furthermore, amygdala neural activity is correlated with a behavioral measure of spatial attention, saccadic reaction times to a barely perceptible target (*Peck et al., 2013*; *Peck and Salzman, 2014*). Correlations between reaction time and amygdala activity have a different sign depending upon the location of the target, with increased activity predicting

shorter reaction times to some locations and longer reaction times to other locations. A framework in which the amygdala merely represents the motivational significance of a stimulus in a valence-nonspecific and spatial-nonspecific manner cannot explain these data due to the spatial dependence of these correlations. If an amygdala neuron merely represents motivational significance then correlations between amygdala activity and reaction times would have the same sign regardless of the location of the saccade target. Increased activity, for example, would predict shorter reaction times for all target locations, which is not consistent with the recent findings.

The reports just described indicate that the amygdala encodes information about space and reward and that neural activity is correlated with spatial attention allocation to stimuli associated with reward. Of course, subjects often exhibit enhanced attention to stimuli threatening aversive events, a behavior that may be mediated by the amygdala as well. In humans, an intact amygdala is vital for guiding gaze towards emotionally-relevant features of fearful face stimuli (*Adolphs et al., 2005*) and for augmenting BOLD responses to these stimuli in the ventral visual areas (*Vuilleumier et al., 2004*) that receive amygdalar input (*Amaral and Price, 1984*) and play an important role in attentional processing (*Reynolds and Chelazzi, 2004*). Moreover, neuroimaging data has revealed that unilateral amygdala lesions are associated with decreased selectivity for negatively valenced stimuli in ipsilateral visual cortices (*Vuilleumier et al., 2004*), a finding consistent with amygdalar projections to visual cortex being primarily ipsilateral (*Iwai and Yukie, 1987*). Thus, the pathway from the amygdala to ventral visual areas may be important in guiding spatial attention towards stimuli associated with both rewarding and aversive outcomes.

If neural activity in the amygdala has a consistent role in modulating spatial attention allocation to both rewarding and threatening stimuli, then amygdala neurons should represent threatening stimuli within a spatial framework, just like they do for rewarding stimuli. Furthermore, neural activity should reflect the allocation of attention to stimuli threatening aversive events. We therefore determined if neural activity in the amygdala reflects the motivational significance of sensory stimuli of both valences in a spatial framework. We recorded the activity of individual amygdala neurons while monkeys performed a task in which stimuli associated with aversive or appetitive outcomes attracted spatial attention. Amygdala neurons represented the spatial location of both reward- and punishment-predicting stimuli, and modulation occurred in the same direction for both types of stimuli. These results suggest that the amygdala provides a means for modulating the neuronal networks responsible for spatial attention allocation to emotionally significant stimuli of both valences.

## Results

### Stimuli associated with appetitive and aversive outcomes attract attention

We trained two monkeys on a detection task in which conditioned stimuli associated with either appetitive or aversive outcomes biased attention (*Figure 1A*). While monkeys maintained fixation, two visual cues appeared briefly for 300 ms on either side of the fixation point. Following cue offset, a variable-length delay period ensued before the 50 ms presentation of a barely-perceptible target in the same location as one of the two cues. The monkeys completed the trial correctly (a 'hit') by making a saccadic eye movement to the location of the target within 600 ms. Generally, 'miss' trials occurred when the monkey failed to make a saccade at all (61% of miss trials) since the timing of target onset was variable and the target itself was difficult to detect; however, miss trials also included those where a saccade was directed towards the location opposite the target (28% of incorrect trials) or elsewhere (11% of incorrect trials). All trials where monkeys' gaze left the fixation window before target onset were repeated such that they could not avoid a particular trial type.

We used three types of cues in our experiments: (1) a reward (R) cue which indicated an opportunity to *obtain* a drop of juice, (2) a punishment (P) cue which threatened delivery of an air puff if the monkey missed the target, and (3) a neutral (N) cue which predicted no outcome for either hit or miss trials (*Figure 1B*). The location of the target was selected randomly, and the reinforcement contingencies enforced on the trial were dictated by the visual cue that had appeared at that location earlier in the trial. Two different cues were randomly chosen to appear on each trial, resulting in 3 randomly interleaved trial types (*Figure 1C*; R/P, R/N, & P/N). The spatial configuration of the cues was also chosen at random on each trial. In addition, two distinct cue sets were interleaved to control for any neural or behavioral preferences specific to a given cue's appearance; the same set of 6 cues was used throughout data collection.

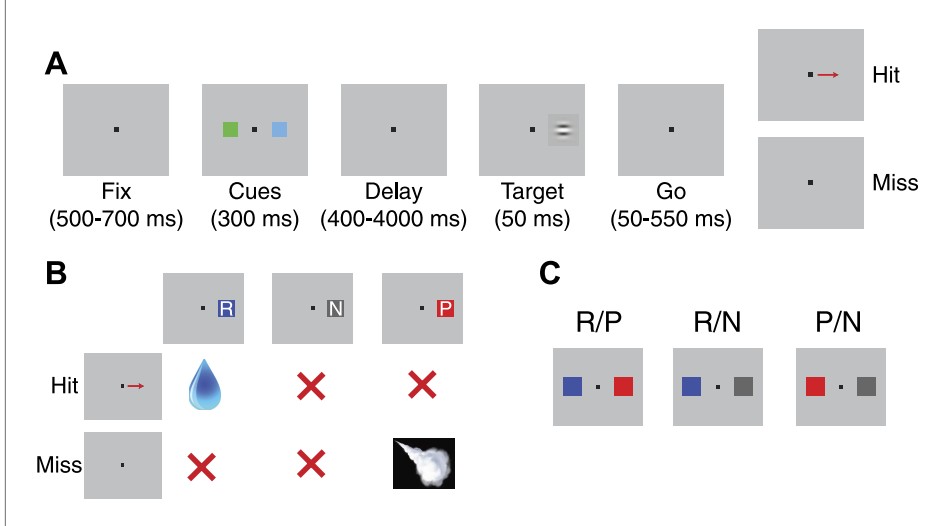

**Figure 1**. Detection task design. (**A**) Task schematic. After fixating, a pair of cues appeared at either side of the fixation point. Following a variable delay, a target appeared at one of the two locations; trials were scored a 'hit' if monkeys made a saccade to the target's location and a 'miss' if they failed to do so. (**B**) Association between cues and outcomes. The table illustrates the outcomes associated with each cue type (given that the target appeared at that cue's location) on hit and miss trials. (**C**) Trial types. On a given trial, monkeys viewed a reward and punishment cue (R/P), a reward and neutral cue (R/N), or a punishment and neutral cue (P/N). Each trial type had two possible spatial configurations (only one configuration is shown).

Behavioral metrics indicate that monkeys understood the reinforcement contingencies dictated by the cues. Monkeys paid more attention to cues that predicted an opportunity to obtain a reward or that threatened a punishment as compared to those that predicted a performance-independent neutral outcome. When the target appeared at the reward cue location relative to either the punishment cue or neutral cue location (R/P & R/N trials), or when the target appeared at the punishment cue location relative to the neutral cue location (P/N trials), hit rate was greater (*Figure 2A*; $\chi^2$-test, $P << 10^{-4}$), reaction time was shorter (*Figure 2B*; Wilcoxon, $p < 10^{-23}$), and false alarm frequency was greater (i.e. frequency of saccades to a cue location *before* the target appeared; *Figure 2C*; $\chi^2$-test, $P << 10^{-4}$). These results were true for each monkey ($p < 0.05$) with the exception that reaction times did not differ on P/N trials for monkey L ($p = 0.45$). Thus, obtaining a reward was of greatest importance for the monkeys, but they also preferred to avoid an air puff rather than responding to a target that resulted in no reinforcement outcome.

## Consistent spatial selectivity for stimuli predicting rewarding or aversive outcomes

During task performance, we recorded the extracellular action potentials of 186 single units (SUA) and 159 multi-unit sites (MUA) from the left amygdala of two monkeys (monkey L: 46 SUA, 45 MUA; monkey O: 140 SUA, 114 MUA). We analyzed firing rates in three time windows: 100–400, 400–700, and 700–1000 ms after cue onset; given that the earliest time of target onset was 700 ms after cue onset, these windows encompassed the cue–driven activity, the delay activity before a target could appear, and the delay activity during which a target could appear, respectively. Firing rates were not analyzed if target onset was before the end of that window. For all forms of selectivity, we used a receiver-operator characteristic analysis (ROC) to compare firing rate distributions across trial conditions, and a Wilcoxon test to assess the significance of firing rate differences ($p < 0.05$). We combined MUA and SUA for all our analyses since the results were similar for each; we address this similarity below with respect to the specific analyses.

Spatial selectivity for the reward cue was characterized by comparing activity from when the reward cue appeared contralateral to the recording site (R-contra trials) to when it appeared ipsilaterally (R-ipsi trials). For this analysis, we combined data from R/P and R/N trials (*Figure 3A*); as we discuss later, neural discrimination between R/P & R/N trials (given a particular spatial configuration) was relatively

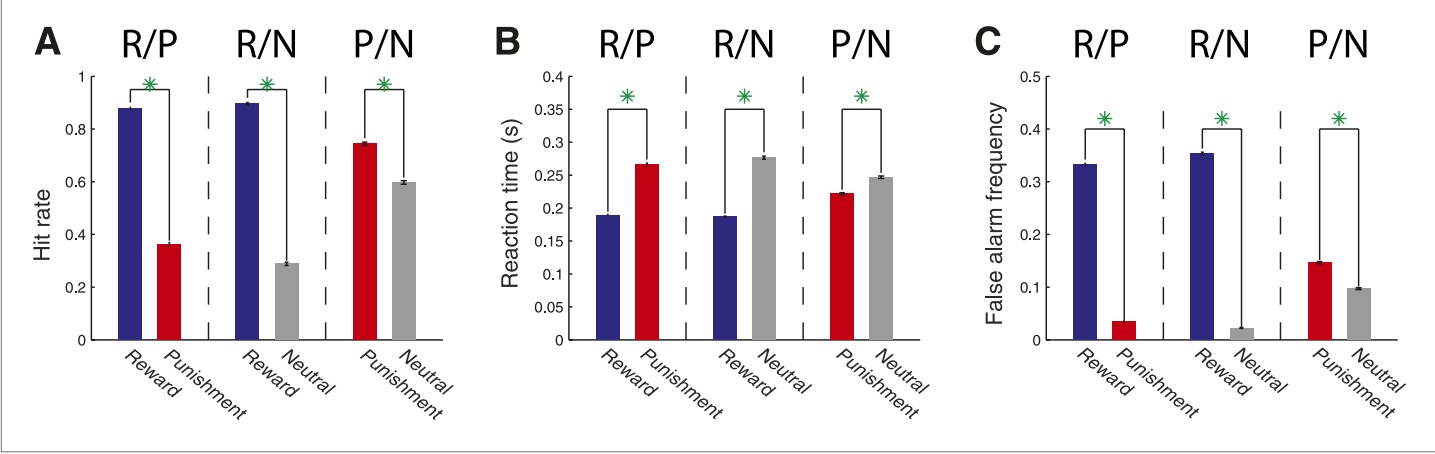

**Figure 2**. Monkeys allocate attention according to stimulus–outcome associations in a detection task. (**A**) Hit rate varied according to the cue–outcome associations. Hit rate is plotted for R/P trials (mean ± standard error; R cue: 0.879 ± 0.004; P cue: 0.362 ± 0.007), R/N trials (R cue: 0.896 ± 0.004; N cue: 0.288 ± 0.006) and P/N trials (P cue: 0.744 ± 0.006; N cue: 0.597 ± 0.007); green asterisks indicate a significant difference between each pair of target conditions ($\chi^2$-test, $p < 10^{-4}$). (**B**) Reaction times for R/P trials (R cue: 0.189 ± 0.001 s; P cue: 0.266 ± 0.002 s), R/N trials (R cue: 0.187 ± 0.001 s; N cue: 0.277 ± 0.002 s) and P/N trials (P cue: 0.222 ± 0.002 s; N cue: 0.247 ± 0.002 s; green astericks: Wilcoxon, $p < 10^{-4}$). (**C**) False alarm frequency for R/P trials (R cue: 0.333 ± 0.003; P cue: 0.034 ± 0.001), R/N trials (R cue: 0.353 ± 0.003; N cue: 0.023 ± 0.001) and P/N trials (P cue: 0.147 ± 0.002; N cue: 0.097 ± 0.002; green asterisks: $\chi^2$-test, $p < 10^{-4}$).

weak compared to the discrimination between R-contra and R-ipsi trials. The location of the reward cue had a strong influence on firing rates such that many sites in each time window responded differentially on R-contra and R-ipsi trials (*Table 1*; *Figure 3B*); this population included sites that had significantly greater (spatial-reward selectivity index >0.5) or lesser (index <0.5) firing rates when the reward cue appeared contralaterally. The overall reward predicted by the cues influenced firing rates as well (*Table 1*; *Figure 3B*); firing rates were often significantly higher (reward selectivity index >0.5) or lower (index <0.5) when the reward cue was presented (R-present trials) than when it was absent (R-absent trials).

We quantified spatial selectivity for the punishment predicting cues by comparing firing rates on P/N trials according to whether the punishment cue appeared contralaterally (P-contra) or ipsilaterally (P-ipsi). We expected that the spatial influence of the punishment cue would be more apparent when it was the primary locus of attention (i.e. when the R cue was absent) and therefore focused our analysis

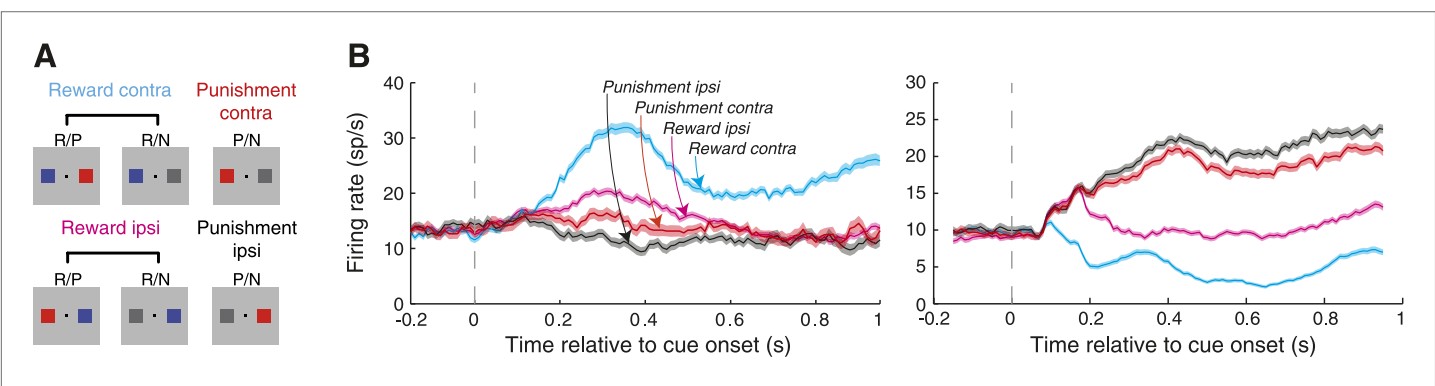

**Figure 3**. The spatial location of cues predicting rewarding and aversive outcomes modulates amygdala neural activity. (**A**) Grouping of trial types for the purpose of neural analyses. (**B**) Example neuron firing rates as a function of time relative to cue onset. Firing rates are plotted for the four trial types (illustrated in **A**) and shading indicates the standard error of firing rates across trials. For each neuron, spatial-reward selectivity was significant in all time windows (Wilcoxon, $p < 0.05$); for the neuron on the left, spatial-punishment selectivity was significant in the first two time windows (100–400 ms, 400–700 ms), and the for the neuron on the right, spatial-punishment selectivity was significant in the last two time windows (400–700 ms, 700–1000 ms).

**Table 1.** Counts of sites with significant selectivity

| | Reward selectivity | Spatial-reward selectivity | Spatial-punishment selectivity |
|---|---|---|---|
| | R present > R absent | R contra > R ipsi | P contra > P ipsi |
| | R present < R absent | R contra < R ipsi | P contra < P ipsi |
| 100–400 ms | 95 (27.5%) | 101 (29.3%) | 26 (7.7%) |
| | 85 (24.6%) | 51 (14.8%) | 22 (6.5%) |
| | n = 345, p < $10^{-4}$ | n = 345, p < $10^{-4}$ | n = 345, p < $10^{-4}$ |
| 400–700 ms | 85 (24.6%) | 67 (19.4%) | 12 (3.6%) |
| | 91 (26.4%) | 64 (18.6%) | 16 (4.8%) |
| | n = 345, p < $10^{-4}$ | n = 345, p < $10^{-4}$ | n = 334, p = 0.0065 |
| 700–1000 ms | 77 (22.5%) | 62 (18.3%) | 8 (2.8%) |
| | 77 (22.5%) | 54 (15.9%) | 9 (3.1%) |
| | n = 342, p < $10^{-4}$ | n = 339, p < $10^{-4}$ | n = 288, p < 0.4969 |

For each cell within the table, the counts of sites with 'positive' selectivity (Wilcoxon, p < 0.05; selectivity index >0.5) and 'negative' selectivity (selectivity index <0.5) are displayed. Percentages were calculated relative to the number of sites for each type of selectivity and time window; this number varied due to the decreasing number of trials available for analysis in later time windows. The frequency of significantly selective neurons, including those with positive and negative selectivity, was tested against chance frequency (Binomial-test, α = 0.05).

on these trials. Sites often fired differentially between P-contra and P-ipsi trial (*Table 1*; *Figure 3B*), with sites firing significantly more (spatial-punishment selectivity index >0.5) or less (index <0.5) on P-contra trials. In agreement with the relatively small effect the punishment cue had on behavior as compared to the reward cue (*Figure 2*), spatial-punishment selectivity was generally weaker than spatial-reward selectivity. The population of spatial-punishment selective sites was significantly smaller than the population of spatial-reward selective sites ($\chi^2$-test, p < $10^{-4}$ for each time window), and the magnitude of spatial-reward selectivity was significantly greater than that of spatial-punishment selectivity (compare |ROC—0.5|; Paired Wilcoxon, p < $10^{-5}$ in each time epoch). These observations do not necessarily imply that the behavioral significance of reward is inherently greater than that of punishment, only that the punishment was made relatively mild in our task in order to keep the monkeys engaged in the task. The relative frequency of sites with positive- and negative-selectivity, on the other hand, was similar for spatial-reward and spatial-punishment selectivity ($\chi^2$-test, p > 0.12 for each time window). Finally, while the frequency of neurons demonstrating significant spatial-punishment selectivity was not greater than chance in the time window where the monkeys could be required to detect the target (*Table 1*; 700–1000 ms), this signal was still apparent for the neural population, which we discuss below.

We next examined the relationship between the types of selectivity that we have described. Consistent with our previous results (*Peck et al., 2013*), we found a strong, positive relationship between reward selectivity and spatial-reward selectivity that was significant in each time epoch (*Figure 4A*; linear regression, p < $10^{-32}$), indicating that those sites that fired more when the reward cue was present tended to fire more when that reward cue appeared contralaterally. Time epoch did not have a significant effect on the slope of these regressions (ANCOVA, p = 0.0940).

Crucially, we next asked whether the spatial selectivity for the reward cue matched the spatial selectivity for punishment cues by examining the linear relationship between spatial-reward and spatial-punishment selectivity indices. Since these two sets of selectivity indices were computed from firing rates on non-overlapping sets of trials (either R-present or R-absent), there was no inherent relationship between the indices and any observed correlation would indicate systematic correspondence of spatial selectivity for reward-predictive and punishment-predictive cues. We also note that an analysis parallel to that in *Figure 4A* with a spatially non-specific form of punishment selectivity (on the x-axis) was not possible since reward had a considerably more profound influence on firing rates.

We observed a clear positive relationship between spatial selectivity for reward and punishment cues in each time epoch (*Figure 4B*; linear regression, p = 1.6*$10^{-22}$, 0.0001, 0.0007 in the 100–400 ms, 400–700 ms, and 700–1000 ms epochs, respectively); these regression slopes were statistically

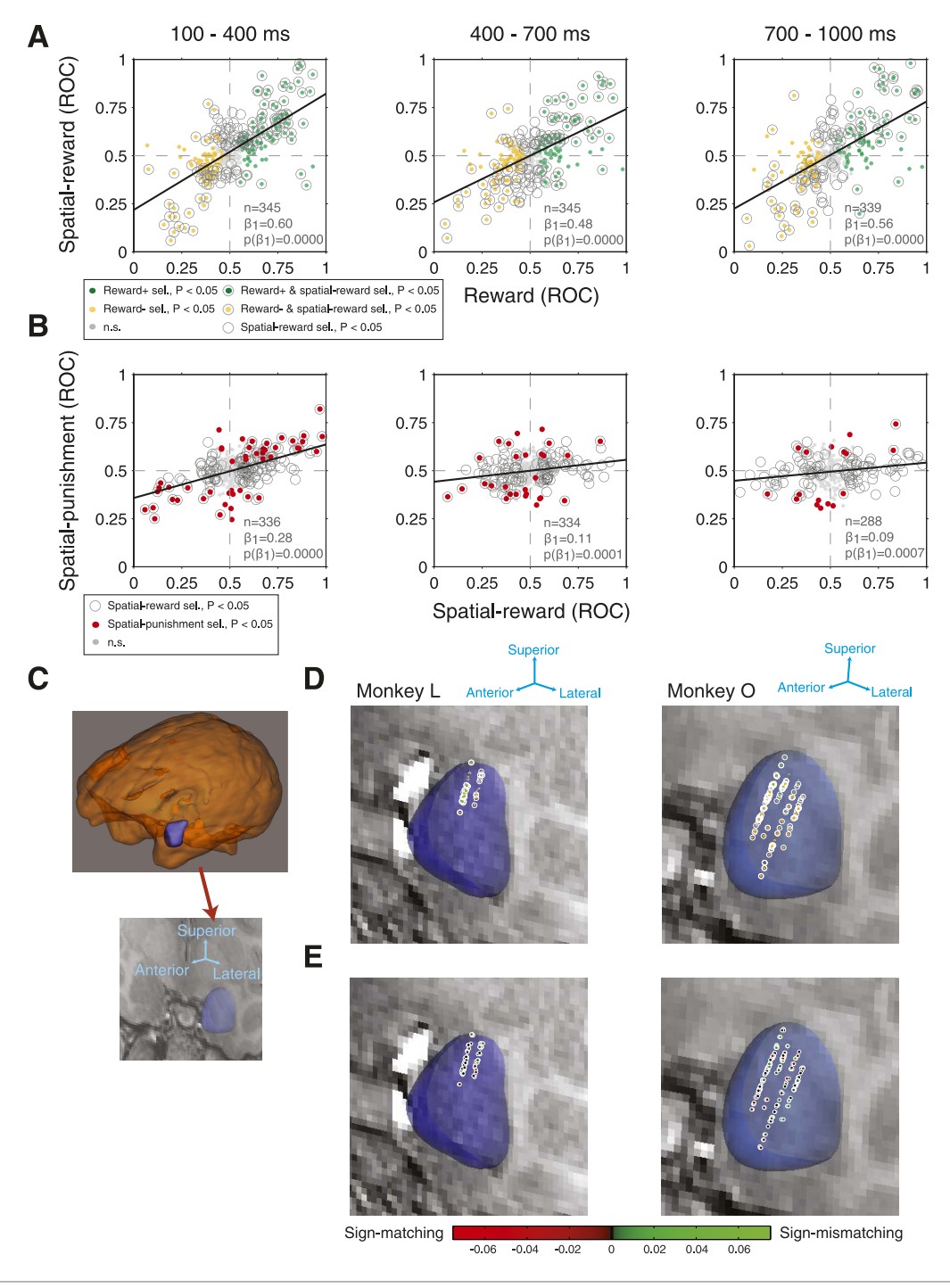

**Figure 4**. Amygdala neurons exhibit consistency between reward selectivity, spatial-reward selectivity, and spatial-punishment selectivity. (**A**) Relationship between reward selectivity and spatial-reward selectivity indices in each time epoch. (**B**) Relationship between spatial-reward selectivity and spatial-punishment selectivity indices. For both (**A**) and (**B**), plot style indicates the significance selectivity for each recording site (see legends) and regressions lines are plotted (significant in each case, p < 0.001). (**C**) 3D reconstruction of the whole brain and the amygdala for Monkey O (top), and the 3D reconstruction of the amygdala overlaid on a single coronal MRI slice for that monkey (bottom). (**D** and **E**) Recording sites. Each coronal slice has been tilted to enable visualization of all electrode tracks. Arrows provide the orientation of the slice after tilting. Each data point represents the location of one site recorded during the task for each monkey (Monkey L: left; Monkey O: right) and the significance of selectivity for that site
*Figure 4. Continued on next page*

*Figure 4. Continued*

(**D**, as in **A**; p < 0.05 in at least one time epoch) or the degree of sign-matching/magnitude of the spatial-reward and spatial-punishment selectivity indices (**E**). In (**E**) positive values (green dots) indicate those neurons with matching signs of selectivity, while negative values (red dots) indicate non-matching selectivity; the brightness of the data points indicates the magnitude of the selectivity.

indistinguishable between monkeys (ANCOVA, p = 0.10, 0.59, 0.74 for each time window) and between SUA/MUA (p = 0.51, 0.59, 0.65). This relationship (*Figure 4B*) was not a byproduct of the correlation between reward and spatial-reward selectivity; using a multiple linear regression, we found that spatial-reward selectivity ($\beta$ = 0.22, 0.12, 0.10 in each time epoch), more so than reward selectivity ($\beta$ = 0.08, −0.00, −0.02), was predictive of spatial-punishment selectivity. There was a significant effect of time epoch on the slope of the regressions lines (ANCOVA, $p < 10^{-6}$), which were greatest in the 100–400 ms epoch (see *Figure 4B*). Despite the difference in the relationship across time epochs, reward, spatial-reward, and spatial-punishment selectivity indices themselves were all positively correlated across time windows (100–400 -> 400–700 ms & 400–700 -> 700–1000 ms; p < 0.0001 except p = 0.14 for spatial-punishment, 400–700 -> 700–1000 ms). The correspondence between spatial-reward and spatial-punishment selectivity was also apparent for individual recording sites. Of those responses (i.e. for each site and time window) with significant spatial-reward and spatial punishment selectivity (n = 64), the sign of selectivity was the same for 54 (84%; Binomial-test, $p < 10^{-7}$). Since attention was biased contralaterally when either a reward (on R/P or R/N trials) or punishment cue appeared contralaterally (on P/N trials), the positive relationship and sign-agreement between these selectivity indices suggest that the spatial signals in the amygdala may influence spatial attention in a similar manner for stimuli promising rewards or threatening punishments.

MRI reconstruction of the recording sites (*Figure 4C–E*) indicated that neurons were not anatomically clustered according to their response selectivity. Sites exhibiting a significant preference for either R-present or R-absent trials were intermingled anatomically, and significant spatial selectivity was widespread (*Figure 4D*). Sites with sign-agreement between spatial selectivity indices (spatial-reward and spatial-punishment indices both >0.5, or both <0.5) were also intermingled with those whose selectivity disagreed in sign (*Figure 4E*).

## Amygdala neurons reflect the push/pull of attention between multiple stimuli

In the previous analyses, firing rates were modulated according to the *primary* focal point of attention, which was either the location of the reward cue on R-present trials (i.e. spatial-reward selectivity) or the punishment cue on R-absent trials (i.e. spatial-punishment selectivity). We next asked if the cue *secondary* in terms of attentional priority modulated neural activity when it was presented simultaneously with the reward cue. Monkeys' behavior indicated that the cue appearing along with the reward cue (either the P or N cue) modulated spatial attention. We compared R/P and R/N trials and found that hit rate was higher when the target appeared at the P cue location on R/P trials than when it appeared at the N cue location on R/N trials (*Figure 5A*; $\chi^2$-test, $p < 10^{-4}$); this effect was similar for both monkeys (monkey O: $p < 10^{-4}$; monkey L: p = 0.0696).

The data presented so far suggests that even when the highly salient R cue is present, monkeys allocate slightly more attention to the 'non-rewarded' field when a P cue appeared there (R/P trials) compared to an N cue (R/N trials). We hypothesized that neural activity would reflect the bias in attention elicited by the P cue. To compare neural activity when the P cue appeared on R/P trials to when the N cue appeared on R/N trials, we again computed selectivity indices (ROC) to assess the difference in firing rates on R/P and R/N trials given a particular spatial configuration. We analyzed the effect of including the P cue contralaterally when the reward cue was ipsilateral (punishment-contra selectivity; *Figure 5B*, top) as well as the effect of the P cue appearing ipsilaterally while the R cue was contralateral (punishment-ipsi selectivity; *Figure 5B*, bottom). On an individual site basis, the P cue had a relatively small influence on firing rate when appearing along with the reward cue. The number of sites that exhibited significant (Wilcoxon, p < 0.05) punishment-contra selectivity was greater than chance only in the 100–400 time window (Binomial-test, p = 0.0011; p > 0.22 otherwise) and was not greater than chance in any window for punishment-ipsi selectivity (p > 0.21). The strong bias in attention towards the reward cue on these trials, made

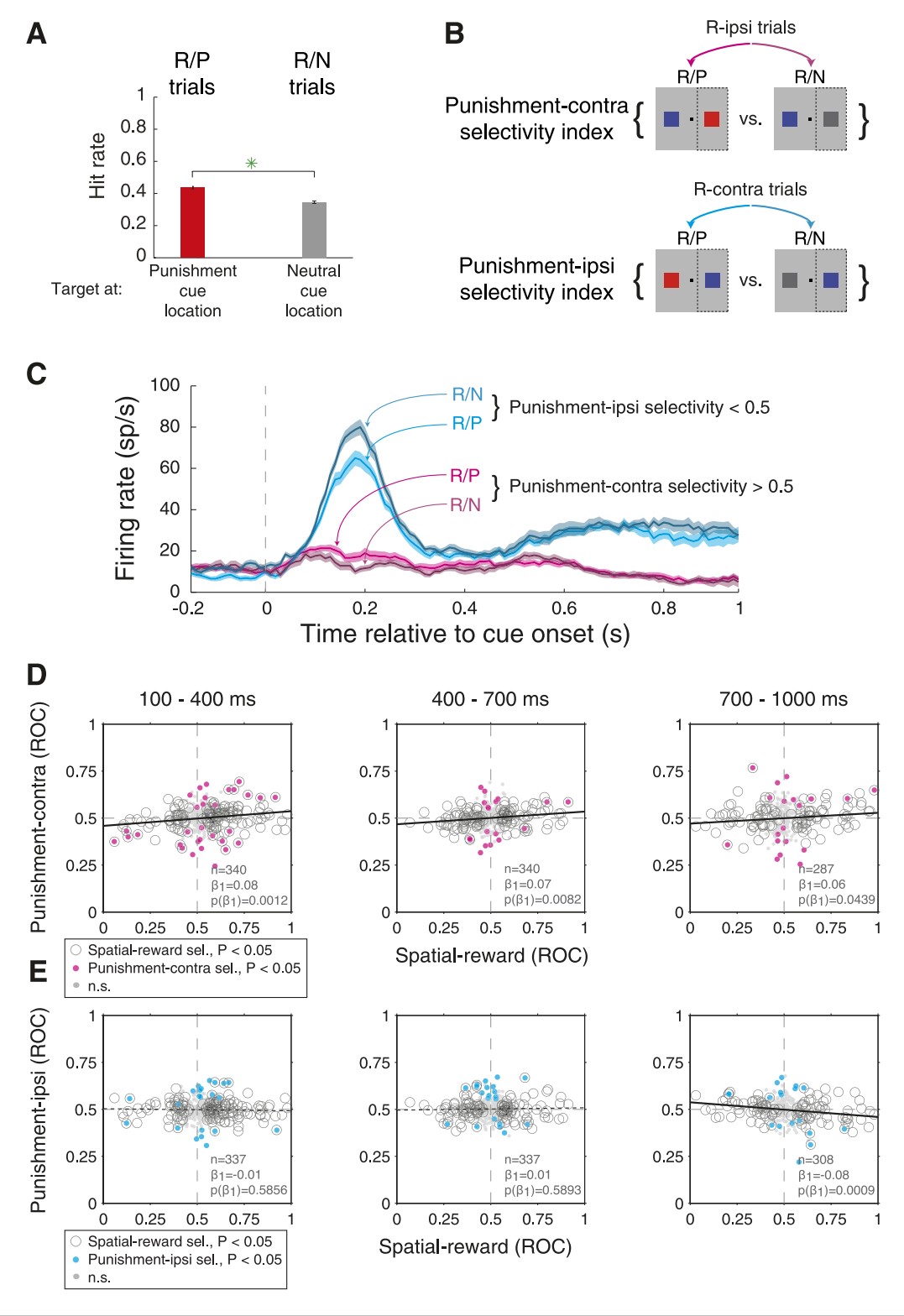

**Figure 5**. Amygdala neurons reflect subtle changes in attention on R-present trials driven by punishment cues. (**A**) Hit rates for secondary cues compared across R/P and R/N trials (bottom). Hit rates were greater for the P cue relative to the N cue; green stars indicate significance (p < 0.05). (**B**) Trial types used to compute punishment-contra and punishment-ipsi selectivity indices and corresponding behavior; dashed rectangles indicate the contralateral

*Figure 5. Continued on next page*

*Figure 5. Continued*

hemifield in each comparison. (**C**) Firing rates of an example neuron on R/P & R/N trials. For this neuron, both punishment-contra and punishment ipsi selectivity were significant (p < 0.05) in the early time window (100–400 ms). (**D** and **E**) Relationship between spatial-reward selectivity indices and (**D**) punishment-contra or (**E**) punishment-ipsi selectivity indices for each time epoch. Spatial-reward selectivity indices on the x-axis are the same as those on the x-axis of *Figure 4B*. Plot style indicates the significance of selectivity indices (see legend); solid and dashed regression lines indicate significant (p < 0.05) and non-significant relationships, respectively.

clear from behavioral metrics (*Figure 2*), likely attenuated neural discrimination between the P cue and N cue.

Despite the relatively weak selectivity of individual sites for these comparisons, modulation of firing rates due to the primary (R) cue predicted the modulation induced by the secondary (P vs N) cue. Consider the example neuron in *Figure 5C*. For this neuron, positive spatial-reward selectivity (selectivity index >0.5) indicated that the neuron fired more when attention was directed contralaterally by the R cue. The punishment-contra selectivity index for this neuron was significantly greater than 0.5 (100–400 ms; Wilcoxon, p < 0.05), indicating that this cell increased firing when more attention was pulled contralaterally by the P cue (relative to the N cue). Moreover, the punishment-ipsi selectivity index was significantly less than 0.5 (100–400 ms; p < 0.05), indicating that the neuron fired less when attention was pulled ipsilaterally. Across recordings, spatial-reward and punishment-contra selectivity indices were positively correlated (*Figure 5D*; linear regression, p = 0.0012, 0.0082, 0.0439 in each time epoch) indicating that those sites firing more when an R cue appeared contralaterally tended to fire more when a contralateral P cue (as opposed to a contralateral N cue) appeared with an ipsilateral R cue; time epoch did not have a significant effect on the slope of these regressions (ANCOVA, p = 0.83).

Strikingly, spatial-reward and punishment-ipsi selectivity indices were *negatively* correlated (as opposed to the positive correlation observed between spatial-reward and punishment-contra indices) in the 700–100 ms epoch (*Figure 5E*; p = 0.0009; p = 0.59, 0.59 in the 100–400 and 400–700 ms epochs, respectively). The difference in this relationship across time epochs was verified by a significant effect of epoch for the spatial-reward/punishment-ipsi relationship (ANCOVA, p = 0.0136). Both punishment-contra and punishment-ipsi selectivity indices were correlated across time epoch (100–400 -> 400–700 ms & 400–700 -> 700–1000 ms; linear regression, p < 0.001 in each case) suggesting consistent coding across time even when its strength differed. Both relationships (*Figure 5D,E*) did not differ significantly between SUA/MUA (ANCOVA, p > 0.15 in each time epoch). Across monkeys, relationships were generally statistically indistinguishable (ANCOVA, p > 0.30) except that the slope between spatial-reward and punishment-contra selectivity indices was significant greater (p = 0.0068) for Monkey L (linear regression, β = 0.23, p = 0.0071) than for Monkey O (β = 0.04, p = 0.12) in the 400–700 ms epoch (*Figure 5D*, center). This discrepancy may be related to the fact that the relationship was more apparent for Monkey O in the 700–1000 epoch (Monkey O: β = 0.06, p = 0.0384; Monkey L: β = 0.03, p = 0.68; *Figure 5D*, right). Since both of these relationships were present late in the trial during times when the target could appear, these signals could have influenced perceptual detection. Overall, amygdala neurons reflect changes in attention driven by stimuli threatening aversive outcomes even when attention is primarily directed at the location of a reward-predicting cue. While not a trial-to-trial measure, the neural tracking of these small biases in spatial attention may be related to the correlation between small trial-to-trial fluctuations in attention and amygdala firing rates that we have described previously (*Peck et al., 2013*; *Peck and Salzman, 2014*).

## The rewarding aspect of avoiding a punishment likely does not account for modulations in attention

The results presented so far demonstrate that stimuli predicting rewards and stimuli threatening delivery of an aversive outcome can enhance attention. Furthermore, amygdala neurons encode spatial information in a manner appropriate for balancing attention between the two hemifields upon viewing stimuli that predict rewarding and aversive events. We next considered the possibility that in our task P cues may attract more attention than N cues because monkeys find the act of avoiding an aversive outcome inherently rewarding, in which case the P cue may not be viewed as being aversive. If this were the case, then both the R cue and P cue could be viewed as more 'rewarding' than the

N cue given the pleasurable possibilities of obtaining reward and avoiding punishment, respectively. Below, we report behavioral measures that confirm that monkeys in fact viewed P cues as aversive stimuli, indicating that the potentially rewarding aspect of punishment avoidance did not confound our neural results.

Two lines of evidence suggest that the P cue retained aversive meaning to monkeys despite its offering the possibility of avoiding the air puff. First, monkeys still experienced the P cue in association with punishment quite frequently during experiments. Overall, monkeys failed to detect 45% of targets appearing at the P cue location (across R/P and P/N trial types); all of these 'miss' trials resulted in air puff delivery. Second, monkeys were more likely to abort trials that included the P cue, indicated that the trials were associated with a less desirable outcome. When the monkey's gaze left the fixation window before the target appeared, some saccades were directed at one of the two cue locations (false alarms) and others were directed elsewhere (aborts). False alarms likely reflect a monkey's desire to detect a target at a given location (*Figure 2C*), but aborts likely indicating that a monkey preferred not to complete a particular trial type. By this logic, if the P cue was associated with aversive meaning to monkeys, we would observe the highest abort rate on the least valuable trials (P/N), and the lowest abort rate on the most valuable trials (R/N).

For each monkey, the inclusion of the R cue tended to decrease the frequency of aborts (*Figure 6*, compare R/P vs P/N trials; Bonferroni-corrected $\chi^2$-test, $p < 10^{-4}$ each for monkey). In contrast, the inclusion of the P cue tended to increase the frequency of aborts (compare R/P vs R/N trials; $p < 0.05$ for each monkey). We observed this behavior during distinct portions of the trial for each monkey: monkey O exhibited this pattern for aborts around the time that the cue was on (0–300 ms after cue onset), and monkey L exhibited this pattern during the subsequent delay (300–1000 ms after cue onset). The results in the *other* time window for each monkey (Monkey L, 0–300 ms; Monkey O, 300–1000 ms) did not contradict these results; neither monkey showed a significant difference in abort frequency between R/P and R/N trials ($p > 0.62$). Abort frequency therefore correlated with the overall reinforcement value of the cues where R/N trials are the most valuable and P/N trials are the least valuable. Overall, these behavioral results suggest that monkeys find the punishment cue aversive, and modulations in attention are therefore unlikely to be due to the rewarding aspect of anticipating punishment avoidance.

## Neural activity is correlated with monkeys' propensity to abort trials

In our task, monkeys have two ways of responding adaptively to the threat of an air-puff: they can detect the target and make an eye movement to avoid air-puff, or they can simply abort the trial by breaking fixation (although the trial would be repeated). We wondered whether neural activity would reflect the threat of the air-puff, and we used the monkeys abort behavior as a behavioral assay of threat. Recall that abort frequency was highest on P/N trials, when reward was not possible but the threat of an air-puff loomed. For each trial type (R-contra, R-ipsi, P-contra, and P-ipsi), we compared

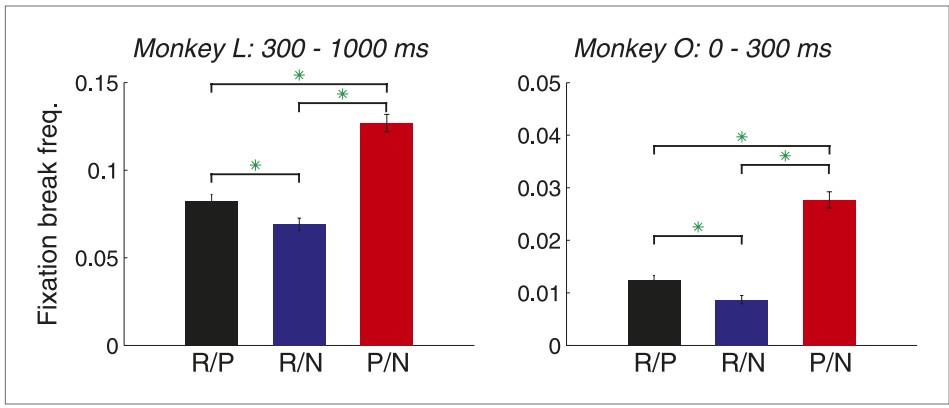

**Figure 6**. Monkeys break fixation in proportion to the reinforcement value of the cues. Fixation break frequency is plotted for each trial type for monkey L (left; mean ± standard error; R/P: 0.082 ± 0.004; R/N: 0.069 ± 0.004; P/N: 0.127 ± 0.005) and monkey O (right; R/P: 0.012 ± 0.001; R/N: 0.009 ± 0.001; P/N: 0.028 ± 0.002). Time windows are relative to cue onset, and green asterisks indicate the significance of comparisons ($p < 0.05$).

firing rates on abort and non-abort trials (Wilcoxon, p < 0.05). Significant selectivity for aborts was more frequent than expected by chance on P-contra trials (7.9%, 7.6%, 10.3% of neurons in each time epoch; Binomial-test, p < 0.0414, 0.0707, 0.0050; *Figure 7A,B*); the proportion of significantly selective neurons was greater than chance in the 100–400 ms epoch for R-contra trials as well (8.2%, p = 0.0469).

We next hypothesized that the neural selectivity for aborts was related to neural signals reflecting the overall value of a trial. We used an ROC analysis to compute 'abort selectivity' indices; indices greater than 0.5 indicated higher firing on abort trials and indices less 0.5 indicated higher firing on non-abort trials. We examined the relationship between abort selectivity indices and reward selectivity indices (as in *Figure 4A*, x-axis). We found a statistically significant negative correlation between reward selectivity indices and abort selectivity indices on P-contra trials in all 3 time epochs (*Figure 7C*,

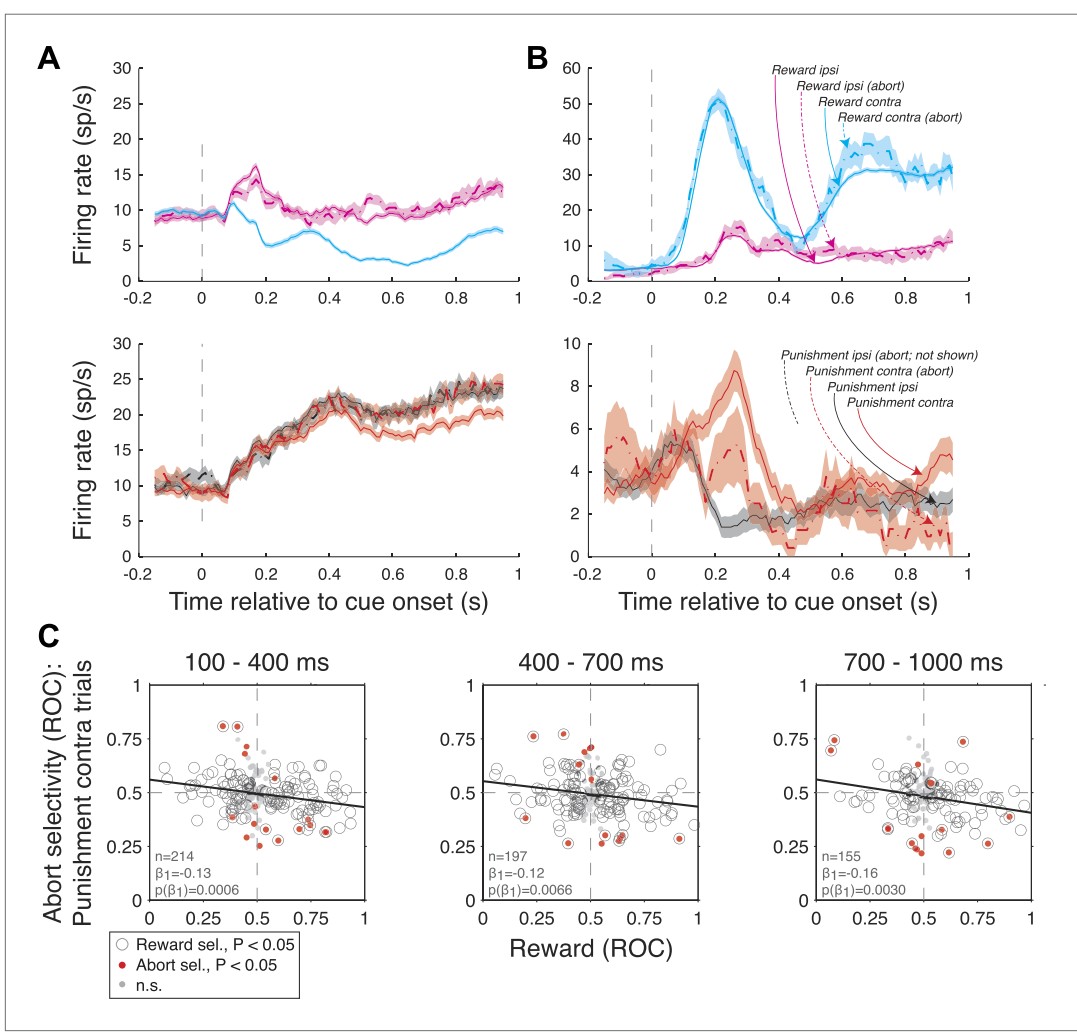

**Figure 7**. Amygdala firing rate selectivity for aborted trials. (**A** and **B**) Example neuron cue-aligned firing rates for aborted (dashed lines) and non-aborted trials (solid lines) on R-present trials (left; R-contra, R-ipsi) and R-absent trials (right; P-contra, P-ipsi). Shading indicates standard error across trials. Note that firing rates for abort trials are not plotted in cases where there were fewer than 6 aborts. (**A**) Neuron with reward selectivity and spatial-reward selectivity indices <0.5 in all time epochs (p < 0.05). Abort selectivity was significant only on P-contra trials in the 700–1000 ms epoch (index = 0.70, p < 0.05). (**B**) Neuron with reward selectivity and spatial-reward selectivity indices >0.5 in all time epochs (p < 0.05). Abort selectivity was significant only on P-contra trials in the 700–1000 ms epoch (index = 0.39, p < 0.05). Different y-axis ranges are used for R-present and R-absent trials to illustrate abort selectivity. (**C**) Relationship between reward selectivity and abort selectivity indices on punishment-contra trials for each time epoch. Plot style indicates the significance of selectivity indices (see legend); solid lines indicate significant regressions (p < 0.05).

linear regression, p < 0.0006, 0.0066, 0.0030 for each time epoch); these relationships were statistically indistinguishable across monkeys (ANCOVA, p > 0.26). Relationships for other trial types and epochs did not approach significance (linear regression, p > 0.11 in each case). Overall, neurons in the amygdala exhibit firing rate selectivity that is predictive of monkeys' assessment of threat, as measured by abort behavior.

## Discussion

Attention to visual stimuli that predict appetitive and aversive events can promote survival by enhancing rapid and accurate approach or avoidance responses in anticipation of upcoming events. Recent work demonstrated that the responses of amygdala neurons are modulated by the spatial location of reward-predictive stimuli (*Peck et al., 2013*), but it remained unknown whether the location of stimuli that threaten aversive experiences influence neural responses in a similar manner. In the present report, monkeys allocated attention towards stimuli that promised reward or threatened punishment. Amygdala neural activity represented the location and motivational significance of visual stimuli in a coordinated manner. Neurons that responded more strongly when a stimulus associated with motivationally significant outcomes appeared also responded more strongly when such stimuli appeared in the contralateral field. Other neurons had the opposite response profile, responding least strongly to visual stimuli associated with motivationally significant outcomes, especially when they appeared in the contralateral field. Importantly, neural activity reflected biases in spatial attention that were induced by the appearance of stimuli associated with appetitive or aversive outcomes. The amygdala therefore may not merely influence attention by modulating arousal, a non-spatial mechanism. Instead, the amygdala may also influence spatial attention—a process distinct from arousal—by encoding information about both the spatial location and motivational significance of sensory cues associated with appetitive and aversive outcomes.

### On the neural representation of valence, motivational significance, and space in the amygdala

A long tradition of work has implicated the amygdala in mediating valence-specific emotional behavior. In rodents, experimental lesions and pharmacological or optogenetic manipulations of neural activity in the amygdala affect approach or defensive behaviors (*Ciocchi et al., 2010*; *Haubensak et al., 2010*; *Stuber et al., 2011*; *Redondo et al., 2014*). Physiological data from rodents and monkeys have linked amygdala neural activity to either appetitive or aversive valence (*Sanghera et al., 1979*; *Nishijo et al., 1988*; *Quirk et al., 1995*; *Schoenbaum et al., 1998*; *Sugase-Miyamoto and Richmond, 2005*; *Paton et al., 2006*; *Belova et al., 2007*; *Shabel and Janak, 2009*; *Bermudez and Schultz, 2010*). Prior work has also implicated the amygdala in mediating valence-nonspecific processes such as autonomic and metabolic arousal (*Davis and Whalen, 2001*). Furthermore, we have previously shown that expectation can modulate amygdala neural responses to rewarding or aversive unconditioned stimuli in a similar manner (*Belova et al., 2007*), reminiscent of unsigned prediction errors in reinforcement learning algorithms (*Pearce and Hall, 1980*; *Schultz and Dickinson, 2000*). These results were suggestive of a role for the amygdala in generating arousal, which can be produced by motivationally significant stimuli of both valences. However, the results did not pertain to visuospatial attention since the observed response characteristics were tied to the unconditioned stimulus, not to conditioned stimuli. Furthermore, in those experiments, conditioned stimuli were not presented at peripheral locations. Recent work has suggested that the amygdala participates in the modulation of spatial attention induced by emotional stimuli (*Vuilleumier et al., 2004*; *Adolphs et al., 2005*; *Peck et al., 2013*; *Ousdal et al., 2014*; *Peck and Salzman, 2014*; *Peck et al., In press*), but these studies have not examined whether amygdala neurons encode spatial information about stimuli of both valences, and, if so, whether that activity correlates with spatial attention deployment.

Scientists have often struggled to disambiguate neural signals related to valence (or value) from those related to attention. Attention can be modulated by motivationally significant stimuli of both valences, and stimuli with high motivational significance may therefore be considered to be more salient. In general, as value increases, motivational significance and attention can increase, presenting an interpretive conundrum (*Maunsell, 2004*). One approach for disentangling these quantities involves testing whether neural responses are modulated similarly for stimuli predicting rewarding and aversive outcomes. Even this approach has caveats, because a neuron might encode both valence and motivational significance. Titration of outcome intensity across valences might be helpful in characterizing the

relative contribution of valence and intensity on neuronal firing. This approach, however, presents experimental challenges because the subjective assessment of stimulus intensity and valence can change within experiments due to satiation to rewards and/or habituation to punishments.

Our prior results indicate that amygdala neurons encode valence when monkeys perform a trace-conditioning task in which behavioral measures of attention were not obtained (*Paton et al., 2006*; *Belova et al., 2008*; *Zhang et al., 2013*). In those studies, neurons preferring positive valence tended to increase firing to a fixation point, which was a mildly positive over-trained visual conditioned stimulus (*Belova et al., 2008*). These positive neurons would then tend to increase firing further if a rewarded conditioned stimulus appeared, or decrease firing if a stimulus associated with an aversive stimulus appeared (*Belova et al., 2008*), suggesting that even though both visual stimuli likely attracted attention, firing rate changed in opposite directions for stimuli associated with reinforcement of different valences. Neurons that preferred negative valence had the opposite response profile.

In the present study, the intensity of rewards and aversive stimuli were likely not equivalent, as rewards had a greater influence on performance. Nonetheless, for a given spatial configuration of stimuli, the presence of a cue threatening an aversive event modulated neural activity in the same direction as a cue promising reward. This could occur if the influence of intensity, or motivational significance, on neural firing was greater than the influence of valence. We did not measure neural responses during a conditioning procedure in these experiments, but the observed differential responses between R-present and R-absent trials bear similarity to valence-related responses observed during trace-conditioning tasks when stimuli were presented over the fovea (*Paton et al., 2006*; *Belova et al., 2008*; *Morrison et al., 2011*).

We observed that amygdala neural activity is similarly modulated by the spatial location of reward- and punishment-predicting stimuli, but monkeys could avoid aversive stimuli by performing the task correctly. Conceivably, this could mean that the modulation of neural activity was related to the rewarding aspects of avoiding an aversive air-puff and not to the threat of an aversive air-puff. For several reasons, we consider it unlikely that the rewarding aspects of punishment avoidance could explain our results. First, air puff delivery was still associated with the P cue on ~45% of trials in which the target appeared at the P location. Second, monkeys aborted R/P trials at a higher rate than R/N trials, suggesting that the punishment cue was more aversive than the neutral cue. Third, prior studies have shown that air puffs have aversive value to monkeys (*Amemori and Graybiel, 2012*), and our experience is that monkeys' propensity to quit performing a task relates to the number of air puffs they have received in that session. Finally, even if monkeys find successful avoidance to be rewarding, this would not be apparent until later in the trial after the monkey successfully acquires the target.

The monkeys' tendency to abort trials more often when an air-puff could occur indicates that the threat of an aversive outcome impacted behavior. Neural activity also reflected the threat of an aversive air-puff, as indexed by whether a monkey aborted a trial. When a P cue appeared on the contralateral side, neural activity in the amygdala predicted whether or not the monkey would later abort the trial in a manner dependent upon the reward selectivity of neurons. Neurons that fire more strongly in a more rewarding situation (e.g. R/N trials) tend to fire less on trials when a monkey breaks fixation. By contrast, neurons that fire more strongly when a monkey is in a less rewarding situation (e.g. P/N trials), fire more on trials when a monkey aborts.

The relationship between the reward selectivity and abort selectivity of amygdala neurons was only apparent on trials in which the P cue appeared contralaterally. It is possible that monkeys exhibit two different kinds of abort behaviors in our task, one in which the monkey's concentration simply lapsed and another where the monkey aborted specifically in response to the value of the trial, that is to the threat of air-puff. The former behavior might be equally likely to occur on all trial types, would not be related to an assessment of threat, and also may have occurred more rarely. The second type of trial abort would occur preferentially on trials in which the R cue did not appear. The fact that amygdala neurons are correlated with trial aborts only when the punishment cue appeared contralaterally, but not ipsilaterally, suggests that there is a spatial component to this response property as well. This is reminiscent of our previous observation that correlations between reaction times and firing rates are apparent only when motivationally significant stimuli appear in the contralateral hemifield (*Peck et al., 2013*). Overall, the relationship between neural activity and the tendency to abort trials does suggest that the amygdala represents threats as well as rewards within a spatial framework.

The current results add an important component to our understanding of neural encoding in the amygdala. Previous studies that described signals in relation to motivational significance or arousal in

the amygdala did not determine whether these properties were represented in a spatial framework (*Belova et al., 2007*; *Shabel and Janak, 2009*). The present study indicates that for many amygdala neurons, the representation of motivational significance is linked to a representation of space. These results highlight the possibility that the amygdala's influence on attention may not be limited to non-spatial processes like emotional arousal. Instead, the amygdala may provide signals that contribute directly to spatial attention. This implies that for a neuron that encodes both space and motivational significance, increases in firing rates may pull attention more towards the contralateral visual hemifield, and less to the ipsilateral field. Since the amygdala also contains neurons with opposite response preferences for space and motivational significance, these other amygdala neurons could have the opposite relationship with spatial attention. Of note, the neural representation provided by the amygdala differs fundamentally from representations that encode motivational significance independent of space, or that encode space and motivational significance but in a non-coordinated manner.

## The representation of motivational significance in other brain structures

Despite a large body of research characterizing the response properties of individual neurons in the primate brain, relatively few experiments have strived to disentangle valence from motivational significance by comparing responses to stimuli predicting either rewards or punishment with those predicting less salient outcomes. In one example, *Kobayashi et al. (2006)* identified a neural population in the lateral prefrontal cortex that responded similarly to stimuli predicting either appetitive or aversive outcomes. Additionally, a population of neurons in the dopamine-producing ventral tegmental area (VTA) and substantia nigra pars compacta (SNc) respond in proportion to the motivational significance of predicted outcomes (*Matsumoto and Hikosaka, 2009*). Finally, *Leathers and Olson (2012)* found that lateral intraparietal area (LIP) neurons fired in proportion to the intensity of predicted outcomes, regardless of valence. Although these neurons may be distinct from those typically recorded in LIP given differences in their fundamental response properties (*Newsome et al., 2013*), they do seem to exhibit responses reflecting motivational significance. The population of neurons described in LIP, however, does not include neurons that prefer less salient stimuli. As a result, LIP does not appear to provide a systematic, coordinated representation of space and motivational significance, as the amygdala does.

Examples of neural responses consistent with the encoding of motivational significance have also been found in the rodent brain. Aside from the report in the amygdala described above (*Shabel and Janak, 2009*), basal forebrain neurons respond according to the motivational significance of outcome-predicting stimuli (*Lin and Nicolelis, 2008*), although the responses described in these studies may have been influenced by the sensory characteristics of the conditioned stimuli themselves. The physiological studies in rodents, however, have not reported a systematic relationship between the encoding of motivational significance and space. In the present paper, some neurons fire more strongly for more motivationally significant (or salient) stimuli, especially when they appear contralaterally. Other neurons fire more strongly for less-salient stimuli, especially when they appear ipsilaterally. To our knowledge, this property has only been described in the primate amygdala.

## The amygdala and the neural control of attention

At least three sets of projections from the amygdala to target structures might influence visual processing and attention. First, the amygdala projects directly to neurons in primate ventral visual areas (*Amaral and Price, 1984*; *Iwai and Yukie, 1987*) whose firing rates modulate depending upon where attention is allocated (*Chelazzi et al., 1993*; *Desimone and Duncan, 1995*). Stimuli predicting aversive events have not been employed while investigating modulation of individual neurons' visual responses in the ventral stream. One prediction of the current work is that if the amygdala directly modulates visual representations, then attention-attracting stimuli associated with aversive outcomes should modulate responses in the same manner as reward-predicting stimuli. Supporting this notion, unilateral amygdala lesions attenuate preferential BOLD response for negative-valence stimuli (*Vuilleumier et al., 2004*).

The amygdala might also influence attention through projections to the basal forebrain or to dopamine neurons. Selective visual attention in rodents appears to involve a projection from the amygdala central nucleus to the basal forebrain (*Holland, 2007*), which may be important for influencing attention-related cortical processing given the basal forebrain's widespread cortical projections (*Mesulam et al., 1983*) and attention-like influences over cortical activity (*Goard and Dan, 2009*). Recent data

indicate that the basal forebrain also encodes spatial and reward information (*Peck and Salzman, 2014*), but the output of basal forebrain neurons does not appear to be correlated with spatial attention on a trial-by-trial basis. Enhanced attention to conditioned stimuli may also involve projections between the amygdala central nucleus to dopamine neurons (*El-Amamy and Holland, 2007*). Amygdala and dopamine neurons are reciprocally connected (*Price and Amaral, 1981*; *Amaral et al., 1982*), and dopamine neurons signal quantities related to motivation (*Matsumoto and Hikosaka, 2009*). It remains unclear whether these pathways might regulate spatial or non-spatial aspects of attention.

The circuit-level mechanisms by which the amygdala might influence attention remain unclear. Distinct populations of amygdala neurons decrease or increase firing rate with respect to spatial attention, unlike modulation in brain areas such as V4 (*Mitchell et al., 2007*) and LIP (*Sugrue et al., 2004*; *Peck et al., 2009*) where firing rates typically only increase when attention is directed towards the contralateral hemifield. The sign of modulation for amygdala neurons does not predict whether spiking statistics are characteristic of excitatory or inhibitory neurons (*Peck et al., 2013*; *Peck and Salzman, 2014*). Amygdala projections to visual cortices are primarily ipsilateral (*Iwai and Yukie, 1987*) and excitatory in nature (*Freese and Amaral, 2006*). Future studies must therefore discern whether the sign of amygdala neurons' spatial selectivity predicts whether projections target excitatory or inhibitory neurons in visual cortex. In addition, much work remains to characterize the spatial specificity of signals provided by the amygdala. The present results suggest that the amygdala could play a role in influencing attention at the level of the hemifield, but the topographical specificity of these signals remains to be characterized.

Human and non-human primates possess a remarkable capacity for dedicating spatial attention to emotionally important visual stimuli ranging from evocative paintings, to wine bottles associated with past pleasures, to frightened faces that reveal looming threats. The physiological data presented in this paper provide a unifying view of the role of the amygdala in representing these emotionally significant stimuli in space so as to modulate attention. Amygdala neurons not only register the emotional significance of stimuli promising reward and threatening aversive events, but they also represent information about the spatial location of those stimuli. Given that the motivational significance and location of stimuli together bias spatial attention, the spatial representation in the amygdala may serve to link our emotional world to cognitive actions—the enhancement of spatial attention to relevant stimuli—that promote our survival.

## Materials and methods

### General methods

Two male rhesus monkeys (*Macaca mulatta,* 8–10 kg) were used in these experiments. All Materials and methods complied with the National Institutes of Health guidelines and were approved by the Institutional Animal Care and Use Committees at the New York State Psychiatric Institute and Columbia University. General methods for these experiments have been described previously (*Peck et al., 2013*).

### Electrophysiology

Recordings from single neurons (SUA) and multi-unit sites (MUA) in the amygdala were made through a surgically implanted plastic cylinder affixed to the skull. Four to eight electrodes were individually lowered into the left (monkeys O and L) amygdala using a multiple electrode microdrive (NaN Instruments, Nazareth, Israel). Extracellular activity was recorded using tungsten electrodes (2 MΩ impedance at 1000 Hz; FHC Inc., Bowdoinham, ME). Analog signals were amplified, bandpass filtered (250–7500 Hz), and digitized (30,000 Hz) for unit isolation (Blackrock Microsystems, Salt Lake City, Utah). We initially defined SUA online as units whose waveforms were clearly distinguishably from noise and/or other units; this classification was made online using either a time-amplitude window or manual clustering in principal component space. After experiments, we reanalyzed the waveforms (Plexon Offline Sorter, Plexon, Dallas, TX) and manually clustered the waveforms of SUA in principal component space; waveform groups were defined as SUA only if they formed a distinct, non-overlapping cluster in principal component space. Because neurons occasionally drift over the course of an experiment such that their waveforms either emerge or descend into the noise cluster, we defined the time interval during which the SUA was clearly distinguishable from the noise and excluded all other data within that session from our analyses. Mutli-unit activity consisted of waveforms that were not sorted as single-units. We re-thresholded the data offline to correct any major deviations in MUA baseline

firing rate due to threshold changes over the course of an experimental session. When MUA and SUA(s) were recorded on the same channel, we removed MUA timestamps within 2 ms of any SUA timestamp to ensure that threshold 'double-crossings' by the single-unit did not contaminate the multi-unit signal.

## Behavioral task

Monkeys performed a detection task designed to assess how reinforcement expectations influenced attention. Each trial began with the presentation of a central fixation point (0.25° × 0.25°); the monkey was required to moved its gaze to a window within 2° of the fixation point. After a fixation period of 500–1500 ms (exponential distribution, λ = 170 ms), two cues appeared at either side of the fixation point along the horizontal axis (7° eccentricity) for 300 ms. Following the offset of the cues, the monkeys continued to fixate during a delay period where no peripheral stimuli were present. At a randomly chosen time 400–4000 ms (exponential distribution, λ = 390 ms) later, a target appeared (50 ms) at one of the two locations at which the cues had been presented. Monkeys were required to make a direct saccade to within 3° of the target between 100–600 ms after its onset.

Eye movements were classified based on where they were directed and when they occurred in the trial. For trials in which the monkeys' eye position left the fixation window before the appearance of the target, eye movements were classified as either a *false alarm* if the saccade was directed within 3° of one of the two cue locations (*Figure 2C*) or as an *abort* if they were directed elsewhere (*Figure 6*). Since the target appeared at a random time, the time range within which false alarms and aborts could occur varied from trial-to-trial. Additionally, saccades within 100 ms of the target were considered to be false alarms or aborts since it is unlikely that the monkey could have reacted to the target's appearance within this short time window. False alarm and abort trials were repeated so that monkeys weren't able to avoid selected trial types; cue configuration, target position, and delay length were re-randomized on repeated trials. For trials where the monkey maintained fixation until target onset, a 'hit' occurred when the monkey successfully made a saccade to the target's location, and a 'miss' occurred when monkeys (1) failed to make a saccade, (2) made a saccade to the opposite cue location, or (3) made a saccade elsewhere. Miss trials that involved a saccade to the opposite cue location or elsewhere were fundamentally distinct from false alarms/aborts since the monkey could receive a punishment in these instances whereas a false alarm or abort simply resulted in a repeated trial. Both hit and miss trials were considered to be 'completed' trials, and outcomes were delivered 1000 ms (monkey L) or 400 ms (monkey O) after trials were completed. Reward consisted of ~1 ml of water controlled by a solenoid and delivered to the monkey through a lick tube; punishments came in the form of a 70 millisecond long 20-40 PSI puff of compressed air aimed at the cheek.

Monkeys learned to associate abstract visual cues with three possible outcomes: reward, punishment, or no outcome. Reward delivery occurred only on hit trials where the target had appeared at the same location as the reward cue. Punishments occurred only on miss trials where the target appeared at the same location as a punishment cue had appeared. These contingencies meant that monkeys never received both a reward and a punishment on the same trial. All completed trials resulted either in reward ('hit' to the target when an R cue had appeared at that location), punishment ('miss' when the target appeared at the location where the P cue had appeared), or no reinforcement ('miss' when the target had appeared at the R location, or 'hit' when the target appeared at the P location). Cues were colored rectangles (2.25 deg.$^2$ at 7° eccentricity) equated for luminance, and we randomly interleaved two distinct sets of cues associated with the same outcomes (6 cues total). Targets were Gabor patches; we adjusted the contrast and size of the Gabors online to maintain an overall performance level of ~70% correct. Because the interval during which the target could appear was long and the reaction time window was relatively short, theoretical chance performance was about 23%. Maximal chance performance levels were determined by assuming that all saccades were made at the specific time in the trial at which a hit was most likely to 'accidently' occur. Given the reaction time window of 100–600 ms, the optimal time to saccade was 100 ms after the 500 ms within which targets were most frequent, in which case ~46% of saccades would occur within the reaction time window. Finally, since these saccades would only be directed at the correct location 50% of the time, chance performance was determined to be 23%. In practice, false alarms were distributed throughout the delay period and occurred most frequently at 790 ms after cue onset, 60 ms *before* the most frequent time of target onset (850 ms), whereas the optimal time to saccade would have been at 1080 ms. Thus, monkeys did not follow this strategy, and effective chance levels were lower than 23%.

## General data analysis

We used two-tailed statistical tests in all instances. Non-parametric Wilcoxon tests were performed on unpaired data (rank-sum test) unless specified otherwise (sign-rank test). Behavioral and neural data was similar across cue sets, so the data were combined except where noted. For selectivity indices, we used a receiver-operator characteristic (ROC) analysis to compare firing rate distributions between conditions; selectivity indices were computed only if at least 15 trials were available for *each* distribution. We used a standard linear regression to assess the relationship between selectivity indices.

Firing rates were analyzed in the time windows 100–400, 400–700, and 700–1000 ms after cue onset. We chose the first window (100–400 ms after cue onset) to capture the visually-driven activity due to the cue; factoring in the ~100 ms visual onset latency of amygdala neurons (*Paton et al., 2006*), this window captures the presentation of the cue, which appeared for 300 ms. The other two time windows (400–700 ms and 700–1000 ms after cue onset) were chosen to surround the earliest time that the target could possibly appear, which was ~700 ms after cue onset. Since monkeys could not predict exactly when the target would appear, it was in their best interest to be prepared for the target's appearance at all times after 700 ms. Since we excluded responses where the target appeared during that time window (46% of targets appeared in the 700–1000 ms window), a decreasing number of trials were available for analysis in later time windows which occasionally results in a site being included in the analysis of early, but not late, time windows. For the neural analysis of abort trials (*Figure 7*), we truncated firing rates at the time that the abort occurred (truncated at the time of the abort, included only if the abort occurred at least 100 ms after the start of the epoch), again resulting in a decreasing sample size for later time windows.

## MRI reconstruction

We logged the inferior/superior, anterior/posterior, and medial/lateral position of each recorded neuron to generate a 3D reconstruction using Brainsight software (*Figure 4C–E*). To determine the degree of sign-matching (*Figure 4E*) for each site's spatial selectivity, we took the product of the spatial-reward and spatial-punishment selectivity magnitude ($|ROC–0.5|$); 'sign-agreement' values were averaged across the three time windows, but similar results are observed if considering any given time window alone. We also estimated the number of recording sites in the basolateral nucleus, the central nucleus, and the anterior amygdala area by comparing our MRI reconstructions with an anatomical atlas (Paxinos, Huang and Toga, 1999 brain atlas). As with previous work (*Peck et al., 2013*), we estimated that the vast majority of our recordings were in the basolateral nucleus of the amygdala (n = 283) as opposed to the central nucleus (n = 51) or anterior amygdaloid area (n = 11). Further, those recordings in the basolateral nuclei excluded the most lateral extent of the amygdala suggesting that these neurons were mainly in the basal and accessory basal nuclei. We targeted these amygdala nuclei because they are the densest source of the projections to the ventral visual stream and/or the basal forebrain, which we believe may have the most direct role in influencing spatial attention.

## Acknowledgements

We thank S Dashnaw for MRI support, G Asfaw for veterinary support, and K Marmon for technical support. We also thank the members of the Salzman Lab for helpful discussion, and Brian Lau for many discussions that aided in the design and analysis of these experiments.

## Additional information

### Funding

| Funder | Grant reference number | Author |
| --- | --- | --- |
| National Institute of Mental Health | R01 MH082017 | C Daniel Salzman |
| National Eye Institute | P30 EY19007 | C Daniel Salzman |
| National Institute on Drug Abuse | R01 DA020656 | C Daniel Salzman |
| National Institutes of Health | T32 HD07430 | Christopher J Peck |
| National Institutes of Health | T32 NS06492 | Christopher J Peck |

| Funder | Grant reference number | Author |
|---|---|---|
| National Institutes of Health | T32 EY139333 | Christopher J Peck |
| National Institutes of Health | T32 MH15174 | Christopher J Peck |

The funders had no role in study design, data collection and interpretation, or the decision to submit the work for publication.

## Author contributions

CJP, Conception and design, Acquisition of data, Analysis and interpretation of data, Drafting or revising the article; CDS, Conception and design, Drafting or revising the article

## Ethics

Animal experimentation: All experimental procedures complied with the National Institutes of Health guidelines and were approved by the Institutional Animal Care and Use Committees at the New York State Psychiatric Institute and Columbia University (protocols 1230 and AAAE4850, respectively).

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
