## [Decision Letter]

Thank you for sending your work entitled “Amygdala neural activity reflects spatial attention towards stimuli promising reward or threatening punishment” for consideration at *eLife*. Your article has been favorably evaluated by Eve Marder (Senior editor), a member of the BRE, and 2 additional peer reviewers.

The Reviewing editor and the other reviewers discussed their comments before we reached this decision, and the Reviewing editor has assembled the following comments to help you prepare a revised submission.

1) The reviewers found you did not clearly lay out the logic and context of the questions/experiments in the Introduction section, and I agree. This is an important issue for the readers. The primary advance you introduce in this study is that amygdala neurons had consistent representations for attention related to both expected reward and expected punishment. This novel observation should be clarified in the text, in particular with respect to what is already known about how both arousal and spatial attention can relate to valence, in general and in terms of the involvement of the amydala.

2) The reviewers suggest that the response epochs should be analyzed separately, and the strength of the relationships between epochs should be presented (Figures 4 and 5). The reviewers also request clarification of whether monkeys knew the precise task timing, in such a way that false alarms could be encoded during the delay periods, for example. Is the task timing reflected in the percentage of hits and false alarm responses?

3) The reviewers noted that the aversive stimulus used in your experiments may not be ideal to understand the relationship between reward and threat. Specifically, you suggest that avoiding an aversive stimulus might be rewarding itself. You have addressed this issue thoroughly with respect to behavior. However, please address this issue more directly with respect to the interpretation of the neural data; that is, does the task design allow you to distinguish between representations of spatial reward and spatial threat in the amydala?

4) Did all analyses include all single and multi-units? Were there any differences in selectivity for the two unit isolation types? What constituted a “single unit”?

5) Figure 2: For RTs, indicate units (presumably seconds). Specify value and error bars for each plot.

---

## [Author Response]

*1) The reviewers found you did not clearly lay out the logic and context of the questions/experiments in the Introduction section, and I agree. This is an important issue for the readers. The primary advance you introduce in this study is that amygdala neurons had consistent representations for attention related to both expected reward and expected punishment. This novel observation should be clarified in the text, in particular with respect to what is already known about how both arousal and spatial attention* can *relate to valence, in general and in terms of the involvement of the amydala*.

We appreciate this comment and have reworked the Introduction of the revised manuscript to better address these issues.

*2) The reviewers suggest that the response epochs should be analyzed separately, and the strength of the relationships between epochs should be presented (*Figures 4 and 5*). The reviewers also request clarification of whether monkeys knew the precise task timing, in such a way that false alarms could be encoded during the delay periods, for example. Is the task timing reflected in the percentage of hits and false alarm responses?*

We now present the neural results separately for each time window and compare findings across time windows in the revised manuscript. The key statistics of these analyses are summarized below:

In all three time windows (100–400, 400–700, and 700–1000 ms after cue onset), we observed a significant, positive relationship between reward and spatial-reward selectivity indices (linear regression, p<10^−32^). Crucially, spatial-reward and spatial-punishment selectivity indices were also positively correlated in each time window (p≤0.0007); these results were statistically indistinguishable between monkeys (ANCOVA, p=0.10, 0.59, 0.74 for each time window, respectively) and between single- and multi-unit data (p=0.51, 0.59, 0.65). To determine how the strength of these relationships varied over time, we used an ANCOVA to compare the regression slopes across time epochs. There was a trend effect of epoch (p=0.0940) for the value/spatial-reward relationship and a significant effect of epoch (p<10^−6^) for the spatial-reward/spatial-punishment relationship. In each case, the relationship was strongest in the 100–400 ms epoch. We also looked at how correlated the selectivity indices themselves were across epoch (100–400 -> 400–700 ms & 400–700 -> 700–1000 ms). Reward and spatial-reward selectivity indices were positively correlated in each case (p<10^−37^); spatial-punishment selectivity indices were positively correlated between the first two time epochs (p=0.0001) but not the last two epochs (p=0.14). These results are included in Figure 4 and the text of the revised manuscript.

A significant, positive relationship between spatial-reward and punishment-contra selectivity indices was apparent in all three time windows (linear regression, p=0.0012, 0.0082, 0.0439) but only in the last time window for spatial-reward and punishment-ipsi selectivity indices (p=0.59, 0.59, 0.0009). This observation was verified by a significant effect of epoch for the spatial-reward/punishment-ipsi relationship (ANCOVA, p=0.0136) but not the spatial-reward/punishment-contra relationship (p=0.83). In each case (Figure 5), relationships were statistically indistinguishable when comparing single- and multi-unit data (ANCOVA, p>0.15). Across monkeys, relationships were generally similar (ANCOVA, p>0.30) except that the slope between spatial-reward and punishment-contra selectivity indices was significant greater (p=0.0068) for Monkey L (linear regression, β = 0.23, p=0.0071) than for Monkey O (β = 0.04, p=0.12) in the 400–700 ms epoch (Figure 5, center); this discrepancy may be related to the fact that the relationship was more apparent for Monkey O in the 700–1000 epoch (Monkey O: β = 0.06, p=0.0384; Monkey L*:* β = 0.03, p=0.68). Finally, both punishment-contra and punishment-ipsi selectivity indices were correlated across time epoch (100–400 -> 400–700 ms & 400–700 -> 700–1000 ms) in each case (p<0.001). These results are included in Figure 5 and in the text of the revised manuscript.

To address whether monkeys were aware of the precise task timing, we examined the frequency of false alarms as a function of time relative to cue onset as well as the target onset frequency, and corresponding hit rates, in each post-cue time bin (Figure 8). False alarms were distributed throughout the trials with peaks occurring while the cues were on the screen and around the time that the target was most likely to appear. Of note, on the assumption that monkeys knew that saccades directed at the cues themselves would never yield an outcome, only false alarms after cue offset were included in the analysis of Figure 2 in the manuscript. Consequently, we examined the timing of the second peak in false alarms (since this was more likely to coincide with the time of target onset). This peak on average occurred slightly before the time when the target was most likely to appear (Monkey *L:* −40 ms; Monkey O: −100 ms). While this result suggests that the monkey may understand the task timing in terms of when the target is most likely to appear, they do not suggest that they followed an optimal strategy in terms of maximizing hit rate. Given the reaction time window (100–600 ms after target onset), the monkeys’ best strategy would have been to saccade 100 ms after the 500 ms period within which targets were most frequent, which was 1070 and 1080 ms after cue onset for Monkey L and Monkey O, respectively (green lines in Figure 8). We also found that the increased false alarm rate did not coincide with increased hit rate. False alarms were most frequent early in the time period where targets could appear, but hit rate grew steadily as a function of time (Figure 8, blue line). We now spell out the logic behind the analyses relevant to these issues in the Materials and Methods section of the revised manuscript.Author response image 1.False alarm frequency (black), target onset frequency (red, normalized), and hit rate (blue, normalized) as a function of time relative to cue onset. Black and red arrows indicate the peaks of the false alarm and target onset frequency, and green arrows indicate the optimal time to saccade that would maximize hit rate. All data was computed in 200 ms windows stepped by 10 ms.

We also determined whether amygdala neural activity predicted whether the monkey would make a false alarm or not. There was only enough data for analyzing the false alarms directed towards the most ‘salient’ location on each trial type (R-cue location on R/P, R/N trials and P-cue location on P/N trials) given that false alarms were more frequent there (Figure 2). For each trial type (R-contra, R-ipsi, P-contra, and P-ipsi), we computed selectivity indices (ROC) comparing firing rates on trials where the monkey false alarmed with those on trials where they did not and evaluated the significance of these firing rate differences (Wilcoxon, p<0.05). The proportion of neurons exhibiting significant selectivity exceeded chance when false alarms were to the contralateral or ipsilateral sides where a reward would have resulted in a correct detection (Binomial-test; contralateral false alarms: 100–400 ms: 13%, p<10^−4^; 400–700 ms: 15%, p<10^−4^; 700–1000 ms: 7%, p=0.1183; ipsilateral saccades: 12%, p<10^−4^; 9%, p=0.0023; 9%, p=0.0045 for the 3 time epochs). On trials where the reward cue did not appear, neural activity did not predict false alarms to a punishment location for either side (p>0.33). Despite the presence of selectivity, this signal did not appear to be coordinated across the population with other aspects of the neurons’ physiological properties; we did not observe a relationship between either spatial-reward selectivity indices or reward selectivity indices (as in Figures 4 and 5) and any of these forms of false alarm selectivity indices (linear regression, p>0.17 in each time epoch). Further, the slope of the regressions between spatial-reward and spatial-punishment selectivity indices (as in Figure 4) did not differ significantly when separately considering false alarm and non-false alarm trials (ANCOVA, p>0.08 in all epochs). Given the lack of a relationship between these findings and other physiological response properties, we find it difficult to interpret the significance of these analyses. As such, we have not included these results in the revised manuscript, but are happy to do so if the reviewers believe that it would make an important contribution.

3) The reviewers noted that the aversive stimulus used in your experiments may not be ideal to understand the relationship between reward and threat. Specifically, you suggest that avoiding an aversive stimulus might be rewarding itself. You have addressed this issue thoroughly with respect to behavior. However, please address this issue more directly with respect to the interpretation of the neural data; that is, does the task design allow you to distinguish between representations of spatial reward and spatial threat in the amydala?

We thank the reviewers for these comments and have expanded upon how these behavioral results support our interpretation of the neural results both in the Results and Discussion section of the revised manuscript. The discussion below includes presentation of a new analysis and result. In particular, we have examined whether neural activity was related to whether the monkey aborted the trial or not. We consider monkeys to have two adaptive ways of responding to the threat of an airpuff. One is to successfully detect the target, which results in avoidance. The second is to abort the trial. Thus, to the extent that neural activity predicts whether a monkey will abort a trial, one might interpret this activity as representing the threat of the airpuff.

The analysis we performed was based on the fact that monkeys’ propensity to abort a trial was correlated with the overall value of trial types (Figure 6).

However, for any given trial type (e.g. P/N trials), monkeys sometimes completed the trial and sometimes aborted it. This variability in behavior could reflect variability in how the monkeys valued a given trial type. We tested the hypothesis that neural activity might reflect this variability in valuation by examining if firing rates predicted whether monkeys would later abort trials. The intuition is that neurons that respond more when a trial is more rewarding (“positive” neurons) should fire less on aborted trials compared to those that were not aborted. By contrast, neurons that respond less when a trial is more rewarding (“negative” neurons) should fire more on aborted trials. One can think of this analysis as being analogous to analyses of choice probabilities in decision-making tasks.

For each trial type (R-contra, R-ipsi, P-contra, and P-ipsi), we used an ROC analysis to compare firing rates on abort and non-abort trials (indices greater than 0.5 indicated higher firing on abort trials and values less 0.5 indicated higher firing on non-abort trials), and a Wilcoxon (p<0.05) to evaluate the significance of firing rate differences. We note that this analysis was only possible for a subset of neurons since aborts were relatively infrequent (Figure 6). Data was analyzed only in epochs prior to or encompassing the abort (truncated at the time of fixation break; we only included data if the abort occurred at least 100 ms after the start of the time epoch being analyzed). As a result, the number of neurons available for analysis decreased for later time epochs. Significant selectivity (Wilcoxon, p<0.05) for aborts was observed in more cells than expected by chance when considering P-contra trials (7.9%, 7.6%, 10.3% of neurons in each of the 3 time epochs analyzed; Binomial-test, p<0.0414, 0.0707, 0.0050). Two example neurons exhibiting this response feature are shown in Figure R2A,B and in Figure 7 of the manuscript. By contrast, the proportion of significantly selective neurons on other trial types was greater than chance only in the 100-400 ms epoch for R-contra trials (8.2%, p=0.0469).

As described above, we hypothesized that abort selectivity indices were related to neural signals reflecting the overall value of a trial, which we characterize using a reward selectivity index (the same index that we used in Figure 4, x-axis). We found a statistically significant negative correlation between reward selectivity indices and abort selectivity indices on P-contra trials in all 3 time epochs (Figure 9 and Figure 7, linear regression, p<0.0006, 0.0066, 0.0030 for each time epoch); these relationships were statistically indistinguishable across monkeys (ANCOVA, p>0.26). Relationships for other trial types and epochs did not approach significance (linear regression, p>0.11 in each case).

In the Discussion section, we briefly speculate about why abort selectivity was limited to trials where the punishment cue appeared contralaterally. One possibility is that monkeys exhibited two different kinds of abort behaviors in our task, one in which the monkey’s concentration simply lapsed and another where the monkey aborted specifically in response to the value of the trial. The former behavior might be equally likely to occur on all trial types. However, for “value-based” aborts, neural selectivity might only be present on trials of more negative value to the monkey, such as the P/N trials. The fact that abort selectivity appears when the punishment cue appeared contralaterally, but not ipsilaterally, suggests that there is a spatial component to this selectivity. This is reminiscent of our previous observation that correlations between reaction times and firing rates are apparent only when motivational significant stimuli appear in the contralateral hemifield (47). We now include these results and possible interpretations in the revised manuscript (Figure 7, Results and Discussion section). Overall, the relationship between neural activity and the propensity to abort trials does suggest that the amygdala represents threats as well as rewards within a spatial framework.Author response image 2.Amygdala firing rate selectivity for aborted trials. (A and **B**) Example neuron cue-aligned firing rates for aborted (dashed lines) and non-aborted trials (solid lines) on R-present trials (left; R-contra, R-ipsi) and R-absent trials (right; P-contra, P-ipsi). Shading indicates standard error across trials. Note that firing rates for abort trials are not plotted in cases where there were less than 6 aborts. (**A**) Neuron with reward selectivity and spatial-reward selectivity indices <0.5 in all time epochs (p<0.05). Abort selectivity was significant only on P-contra trials in the 700-1000 ms epoch (index = 0.70, p<0.05). (**B**) Neuron with reward selectivity and spatial-reward selectivity indices >0.5 in all time epochs (p<0.05). Abort selectivity was significant only on P-contra trials in the 700-1000 ms epoch (index = 0.39, p<0.05). Different y-axis ranges are used for R-present and R-absent trials to illustrate abort selectivity. (**C**) Relationship between reward selectivity and abort selectivity indices on punishment-contra trials for each time epoch. Plot style indicates the significance of selectivity indices (see legend); solid lines indicate significant regressions (p<0.05).

4) Did all analyses include all single and multi-units? Were there any differences in selectivity for the two unit isolation types? What constituted a “single unit”?

All of our analyses of population neural activity (Figures 4 and 5 and Table 1) included both single-unit activity (SUA) and multi-unit activity (MUA). For clarity, we now mention this in the beginning of the Results section.

The positive relationship between reward and spatial-reward selectivity indices (Figure 4) was significant for both SUA and MUA (p<10^−5^ in each time epoch) although the regression slopes were greater for SUA in the 400–700 and 700–1000 ms epochs (SUA: β = 0.53, 0.60; MUA: β = 0.30, 0.41; ANCOVA, p=0.0136, 0.0491). The positive relationship between spatial-reward and spatial-punishment selectivity indices (Figure 4) did not differ between SUA/MUA in any epoch (ANCOVA, p>0.50) and was generally significant for each (p<0.05 in each epoch, except p=0.2374 in the 700–1000 ms epoch for MUA). We also observed a similar selectivity for SUA and MUA in the analyses of Figure 5; the slope of the regression lines did not differ between activity type for either comparison (ANCOVA, p>0.15).

We have now expanded the Discussion of our criteria for defining SUA and MUA. We initially defined SUA online as units whose waveforms were clearly distinguishable from noise and/or other units; this classification was initially made online using either a time-amplitude window or manual clustering in principal component space. After experiments, we reanalyzed the waveforms using Plexon Offline Sorter and manually clustered the waveforms of SUA in principal component space; waveform groups were defined as SUA only if they formed a distinct, non-overlapping cluster in principal component space. Because neurons occasionally drift over the course of an experiment such that their waveforms either emerge or descend into the noise cluster, we defined the time interval during which the SUA was clearly distinguishable from the noise and excluded all other data within that session from our analyses of SUA.

*5)*
Figure 2*: For RTs, indicate units (presumably seconds). Specify value and error bars for each plot*.

We apologize for not including the units for reaction times (seconds), and have updated Figure 2 of the revised manuscript. We have also included the mean and standard error for all the data in Figures 2 and 6 in the legends of each figure.